# Separation of α-Lactalbumin-Enriched Fractions from Caprine and Ovine Native Whey Concentrate by Combining Membrane and High-Pressure Processing

**DOI:** 10.3390/foods12142688

**Published:** 2023-07-12

**Authors:** María Romo, Massimo Castellari, Ricard Bou, Pere Gou, Xavier Felipe

**Affiliations:** 1Food Processing and Engineering Programme, Institute for Food and Agricultural Research and Technology (IRTA), Granja Camps i Armet s/n, Monells, 17121 Girona, Spain; maria.romo@irta.cat (M.R.); pere.gou@irta.cat (P.G.); 2Food Safety and Functionality Programme, Institute for Food and Agricultural Research and Technology (IRTA), Granja Camps i Armet s/n, Monells, 17121 Girona, Spain; massimo.castellari@irta.cat (M.C.); ricard.bou@irta.cat (R.B.)

**Keywords:** HPP, β-Lactoglobulin (β-Lg), α-Lactalbumin (α-La), whey, proteins, goat, sheep

## Abstract

Whey from goat and sheep have been gaining attention in the last few years for their nutritional properties. Unfortunately, β-Lg, not found in human milk, may trigger infant allergies if used in infant food formulations, so there is a growing interest in developing ingredients derived from whey with higher α-La/β-Lg ratios. The objective of this work was to study the effect of high-pressure processing (HPP) on caprine and ovine native whey concentrates (NWC) in order to obtain α-Lactalbumin (α-La)-enriched fractions. NWCs were treated at 600 MPa (23 °C) for 2, 4, and 15 min and two pH conditions were studied (physiological pH and pH 4.60). The concentration of β-Lg in supernatant fraction after HPP significantly decreased after 2 min of treatment, while the concentration of α-La was unchanged in both goat and sheep samples. Longer HPP processing times (up to 15 min) progressively increased α-La purification degree but also decreased the α-La yield. Caprine and ovine NWCs treated at physiological pH provided better α-La yield, α-La purification degree, and higher β-Lg precipitation degrees than the corresponding acidified samples, while the corresponding NWC supernatant (NWC_sup_) showed lower values for both surface hydrophobicity and total free thiol indices, suggesting a higher extent of protein aggregation. Effects of sample acidification and the HPP treatment were opposite to those previously reported on bovine NWC, so further characterization of caprine and ovine β-Lg should be carried out to understand their different behavior.

## 1. Introduction

Goat and sheep contribute only marginally to global milk production (3.45% vs. 81.08% in the case of cow [1]), but they play an important role in the economy of Mediterranean and Middle Eastern countries, providing a valuable alternative to bovine milk. The nutritional, functional, and organoleptic properties of goat and sheep milk turn them into a potential solution to the increase in consumer demand for healthier food products seen in recent years [2].

Many studies have highlighted the beneficial effects of goat and sheep dairy products on human health, being considered more easily digestible than cow milk [3,4]. Caprine milk has been described as an alternative for infants with cow milk allergies on account of its nutraceutical and hypoallergenic properties due to its high digestibility and the lower presence of β-Lactoglobulin [5,6].

Sheep and goat milk are mainly used to elaborate cheeses; the high amounts of cheese whey generated as a side-product of this process are generally processed for animal nutrition applications, even if their protein fraction is also of particular interest for human nutrition, since these whey proteins (WPs) have been demonstrated to be a source of bioactive peptides with multiple applications, such as antimicrobial or antihypertensive activities [7,8].

As in other ruminant species, α-Lactalbumin (α-La) and β-Lactoglobulin (β-Lg) are the most represented whey proteins (WPs) in goat and sheep milk, with a total concentration varying from 1.27 to 3.07 g/L in goat milk and from 0.95 to 5.97 g/L in sheep milk [9]. These WPs show high homology with the bovine ones, showing minimum changes in their amino acid sequences [6,7].

Unfortunately, β-Lg, which is not found in human milk, may trigger infant allergies when WPs concentrates (WPC) and isolates (WPI) are used in infant food formulations, so there is a growing interest in developing ingredients derived from whey with higher α-La/β-Lg ratios for these valuable applications [10,11,12].

Several whey up-cycling processes have been suggested to obtain whey fractions enriched with α-La, including membrane filtration [13,14,15], selective precipitation [16,17], chromatographic separation [18,19,20], and hydrolysis [21,22]. The scale-up of these technologies at the industrial level could be challenging due to different causes including (i) long processing times; (ii) low selectivity; (iii) excessive cost; and (iv) the need of additives [23].

More recently, high-pressure processing (HPP) has been suggested as an alternative technology to obtain an α-La-enriched fraction from cow whey by taking advantage of the differential effects of pressure on the structure of WPs. Fractionation of WPs is based on the irreversible denaturation of β-Lg which starts at around 300 MPa, while α-La maintains its native conformation up to 500 MPa [24,25,26]. Felipe et al. [27] confirmed the different baroresistance of α-La and β-Lg in caprine milk, observing a significant decrease in β-Lg solubility after 10 min at 250 MPa, while α-La solubility was affected only under stronger conditions (>500 MPa). More recently, other authors found a similar behavior of the WPs after treating caprine milk in the range of pressure between 200 MPa and 500 MPa [28,29]. Likewise, in ovine milk, Huppertz et al. [30], Moatsou et al. [31], and Sakkas et al. [32] noted that α-La was denatured at pressures higher than 600–650 MPa, while β-Lg started to be affected from 200 MPa. Although fractionation of α-La-enriched fractions from bovine whey by HPP have been described in some papers [25,33,34,35], to the best of our knowledge, there is still a lack of studies focused on the HPP of ovine and caprine concentrated native whey to obtain α-La-enriched fractions.

Thus, the aim of this work was to study the effect of HPP treatments on the main proteins (α-La and β-Lg) in ovine and caprine native whey concentrate (NWC) obtained via micro and ultrafiltration at pilot plant level without any heat treatment. A combination of concentration and HPP treatment was explored as a potential approach for future industrial scale-up in order to obtain high-value α-La-enriched fractions. In order to obtain a suitable α-La-enriched fraction, several combinations of pH, initial sample, pressure, and HPP processing time were assessed.

## 2. Materials and Methods

### 2.1. Native Concentrated Whey Preparation and High-Pressure Processing (HPP)

One lot of raw skimmed goat milk and one lot of raw sheep milk (500 L each) were supplied by local farms. Raw milk was microfiltered and ultrafiltered following the same protocol as in Romo et al. [35], using a SW40 MMS AG and a SW18 filter systems (MMS AG Membrane Systems, Urdof, Switzerland) according to Figure 1. When the native whey was obtained and concentrated, sodium azide (0.33%) was added and stored at 4 °C for 12 h until the high-pressure processing treatment (HPP) was complete.

The main compositional parameters of the resulting native whey concentrates (NWCs) are summarized in Table 1. Fat content (% *w/w*), protein content (%), total dry matter, and ash (% *w/w*) were determined according to Romo et al. [35], following the protocols ISO 1211/IDF1 [36], ISO 8968-3/IDF20-3 [37], ISO 2920:2004/IDF58:2004 [38], and BOE-A-1977-16116 [39], respectively. Lactose was estimated by subtracting the total fat, total protein, and ash from the total dry matter, while pH was determined with a pH-meter (sensION+ PH3, HACH Co., Loveland, CO, USA).

NWC from both animal sources were processed at two pH conditions: physiological pH (P-pH) and acidified (pH 4.6); performing the acidification with 1 M HCl. Samples (50 mL), equilibrated at room temperature, were transferred in high-density polyethylene (HDPE) bottles (Nalgene, Thermo Fisher Scientific Inc., Whaltham, MA, USA) for HPP processing. HPP processing was carried out at 600 MPa for 2, 4, and 15 min at room temperature (23 °C), in agreement with the results observed in NWC from cow [35], with n = 3 independent treatments for each of the two NWCs. Untreated samples for both pH conditions were used as controls. The HPP was performed in a Wave6000/120 industrial equipment (Hiperbaric, Burgos, Spain), where water was used as the pressure transmission medium. The compression and decompression rates were 150 MPa/min and <2 s, respectively, according to the data obtained from the SCADA software.

After HPP processing, the samples were centrifuged (3270× *g*, 4 °C, 20 min) (Beckman Avanti^®^ JXN-30, Beckman Coulter, Inc., Brea, CA, USA). All pellets and supernatants (NWC_sup_) were weighed, and density was considered for the calculation of the supernatant volume. The HCl volume was also considered in the case of the acidified samples.

### 2.2. Chemicals and Standards

Acetonitrile of HPLC grade, trifluoracetic acid (TFA), hydrochloric acid (HCl), 2-mercaptoethanol, sodium citrate, urea, 8-anilino-1-naphtalenesulphonic acid (ANS) ethylenediaminetetraacetic acid equivalent (EDTA), dibasic potassic phosphate, 5,5′-dithio-bis-(2-nitrobenzoic) acid (DTNB), bovine serum albumin (BSA), DL-dithiothreitol (DTT), 2,2-Bis(hydroxymethyl)-2,2′,2″-nitrilotriethanol (Bis-Tris), and sodium azide were provided by Sigma Aldrich (Sigma-Aldrich^®^ Merck, Darmstadt, Germany). Reference standards of bovine whey proteins (α-La and β-Lg A and β-Lg B isoforms) were provided by Cerilliant (Sigma-Aldrich^®^ Merck, Darmstadt, Germany).

### 2.3. HPLC Proteins Quantification

Analysis of whey proteins (α-La, β-Lg A, β-Lg B) was carried out following Marciniak et al. (2018) with small adjustments. A Luna^®^ 5 μm C18_(2)_ column (150 × 4.6 mm) (Phenomenex^®^, Torrance, CA, USA), a binary HPLC pump 1525 equipped with a 717 plus Autosampler, and a photodiode array detector 2996 (Waters, Milford, MA, USA) were used.

The pH of NWC_sup_ samples was adjusted to 4.6 with 1 M HCl. Samples were diluted 1:1 (*v/v*) with buffer (0.1 M Bis-Tris, 0.3% 2-mercaptoethanol, 5.37 mM sodium citrate), and further diluted 1:500 (*v/v*) with a mixture of TFA, water, and acetonitrile (1:619:380 *v/v*). Diluted samples were filtered directly in the vial through a 0.45 mm cellulose acetate syringe filter (Scharlab, S.L., Barcelona, Spain). The 50 μL sample was injected. The column temperature was set at 35 °C and flowrate at 1.1 mL·min^−1^.

Chromatographic separation was performed with a gradient elution between mobile phase A (0.1% *v/v* TFA in water) and B (0.1% *v/v* TFA in acetonitrile) by varying linearly the percentage of mobile phase B from the initial 30% to 45% in 26 min. The system was controlled by Empower Pro v. 2 software (Waters, Milford, MA, USA). Quantification was performed at 214 nm, and proteins were identified by comparing retention time and ultraviolet (UV) spectra of the peaks with those of the standard compounds. For the quantification, calibration curves with different amounts of the external commercial standards were created.

### 2.4. Process Performance Parameters

After the HPP treatment and centrifugation, two fractions were obtained, i.e., an insoluble white precipitate and the supernatant (NWC_sup_) which was enriched with α-La. As the supernatant was a fraction of the initial NWC sample (Appendix A), its volume was considered to calculate the overall yield of the process.

The main process performance parameters were calculated for NWC_sup_ according to the following equations:(1)$$\alpha -\mathrm{La}\;\mathrm{Yield}\;(\%)\;=\frac{[\alpha -La{]}_{s}\cdot V_{s}}{[\alpha -La{]}_{c}\cdot V_{c}}\;\cdot \;100$$
(2)$$\alpha -\mathrm{La}\;\mathrm{Purification}\;\mathrm{degree}\;(\%)=\frac{[\alpha -La{]}_{s}}{[\alpha -La{]}_{s}+[b-LgA{]}_{s}+[b-LgB{]}_{s}}\;\cdot \;100$$
(3)$$\beta -\mathrm{Lg}\;A\;\mathrm{precipitation}\;\mathrm{degree}\;({\mathrm{PR}}_{\beta -\mathrm{Lg}\;A})\;(\%)=100-\frac{[\beta -LgA{]}_{s}\cdot V_{s}}{[\beta -LgA{]}_{c}\cdot V_{c}}\;\cdot \;100$$
(4)$$\beta -\mathrm{Lg}\;B\;\mathrm{precipitation}\;\mathrm{degree}\;({\mathrm{PR}}_{\beta -\mathrm{Lg}\;B})\;(\%)=100-\frac{[\beta -LgB{]}_{s}\cdot V_{s}}{[\beta -LgB{]}_{c}\cdot V_{c}}\;\cdot \;100,$$
where

[α-La]_s_ = concentration of α-La in the supernatant after the HHP treatment;

[α-La]_c_ = concentration of α-La in the corresponding control sample;

[β-Lg A]_s_ = concentration of β-LgA in the supernatant after the HHP treatment;

[β-Lg A]_c_ = concentration of β-LgA in the corresponding control sample;

[β-Lg B]_s_ = concentration of β-LgB in the supernatant after the HHP treatment;

[β-Lg B]_c_ = concentration of β-LgB in the corresponding control sample;

V_s_ = volume of the supernatant recovered after the HHP treatment;

V_c_ = volume of the sample before the HHP treatment.

### 2.5. Hydrophobicity Index

The hydrophobicity index (%) was determined according to Steen et al. [40] with small modifications. Dibasic potassic phosphate buffer (0.05 M, pH 7.40) was used for diluting the NWC_sup_ samples (0.16, 0.08, 0.04, 0.02, 0.01, 0.005, and 0.0025 *v/v* %), and 4 mL of each dilution was incubated with 20 mL of ANS 8 mM for 10 min in darkness. Fluorescence (λ_ex_ = 390 nm, λ_em_ = 480 nm) was measured in a Varioskan Flash (Thermo Fisher Scientific, Vantaa, Finland).

The hydrophobicity index was calculated as a percentage of the unprocessed sample (P-pH) according to Equation (5), where Kd is the hydrophobicity dissociation constant, obtained by plotting the measurement of each dilution vs. the concentration and calculating the slope of the linear regression.
(5)$$\mathrm{Hydrophobicity}\;\mathrm{Index}\;(H_{i})\;(\%)=\frac{K_{dsample}}{K_{dcontrol}}\;\cdot \;100$$

### 2.6. Total Free Thiol Groups Index

The total free thiol group index (TFTI) was estimated according to Yong-Sawatdigul and Park [41]. A total of 1 mL of NWC_sup_ was mixed with 9 mL of buffer (dibasic potassic phosphate 0.05 M, urea 8 M and EDTA 10 mM). A total of 4 mL of the previous mixture was incubated with 0.4 mL of 0.1% DTNB for 25 min at 40 °C; then, the absorbance was measured at 412 nm with a Varioskan^®^ Flash (Thermo Fisher Scientific, Vantaa, Finland). Different solutions of a BSA standard (in the range 0–8 mg/mL) were used to create a calibration curve. The results were expressed in BSA equivalents (mg/mL) and then normalized considering the corresponding HPP-untreated sample at physiological pH, following Equation (6).
(6)$$\mathrm{Total}\;\mathrm{free}\;\mathrm{thiol}\;\mathrm{groups}\;\mathrm{Index}\;(\mathrm{TFTI})\;(\%)=\frac{{TFT}_{Isample}}{{TFT}_{Icontrol}}\;\cdot \;100$$

### 2.7. Statistics

All statistical analyses were performed in triplicate using JMP v. 16.2.0 software (SAS Institute Inc., Cary, NC, USA). ANOVA and Tukey’s honest significant difference (HSD) were carried out to study the effect of independent variables on the dependent variables (statistical significance set at *p* < 0.05). A *t*-test was performed to check significant differences between untreated and HPP-treated samples for the α-La yield, the β-Lg A and β-Lg B precipitation degrees, the hydrophobicity index, and the total free thiol group index.

## 3. Results and Discussion

The effect of high-pressure processing (HPP) on ovine and caprine NWC at physiological and acidified pH was studied by applying process conditions (600 MPa and 23 °C), which provided the best balance between α-La yield and purification degree in previous works with bovine whey [34,35].

### 3.1. Main Process Performance Parameters

Acidification before HPP processing increased the concentration of α-La in NWC_sup_ of both sheep and goat untreated samples (Figure 2).

Short HPP treatments (2 and 4 min) did not have a significant effect on the concentration of α-La in ovine and caprine NWC_sup_ at both pH conditions (acidified and P-pH) in comparison to the corresponding untreated samples.

α-Lactalbumin concentration was only significantly reduced when ovine NWC was treated for fifteen minutes at acidic and physiological pH. No significant decrease was observed in goat NWC_sup_ for any of the studied HPP conditions (Figure 2, Appendix A). These findings agree with the reported α-La baroresistance from ovine [30,31] and caprine milk [27] treated under similar conditions.

On the contrary, the concentration of β-Lg was significantly reduced by HPP processing in both, goat and sheep NWC_sup_. The behavior of the A and B forms of β-Lg in sheep was very similar (see Appendix A), so all the performance parameters are shown considering the sum of the A + B forms. Precipitation of β-Lg was already significant after 2 min, and it increased with longer processing times in acidified samples. In addition, the effect of HPP on β-Lg was more evident in samples treated at physiological pH, where this protein was undetectable after fifteen-minute HPP treatments (Figure 3, Appendix A).

β-Lactoglobulin precipitation degree (Pre_β-Lg_) was higher than 95% for 2-min HPP processing times and was higher than 99% for 15-min HPP processing times in both goat and sheep samples treated at physiological pH (Figure 4). In both species, β-Lg precipitated significantly faster at physiological pH than in samples acidified at pH 4.6. Incomplete precipitation of β-Lg forms in both goat and sheep acidified samples (Pre_β-Lg_ of 89.93% and 87.50%, respectively) was observed even after a 15-min HPP processing time. Furthermore, for short HPP processing times (2–4 min), β-Lg seemed to be more prone to precipitate in acidified sheep NWCs than in goat ones (Pre_β-Lg_ of 68.98–71.71 and 23.82–49.23%, respectively). Under our conditions, β-Lg precipitation degree at physiological pH was higher than reported in ovine milk [30,31] and in caprine and ovine milk [28,42] that were not concentrated; these results could be partially due to the high protein content in NWCs, which increased the interaction between the β-Lg molecules and their precipitation. Overall, sample acidification had a contrary effect on Pre_β-Lg_ than that reported of bovine NWCs [35].

As a general rule, α-La yield (Figure 5) progressively decreased with processing time in both goat and sheep NWC_sup_. It seemed that, for short processing times, the α-La yield could be higher in goat than in sheep (75.6 and 64.5% for goat vs. 62.0 and 44.8% for sheep, at 2 and 4 min, respectively), but after fifteen minutes of HPP treatment, the values of this parameter were similar for the two species. For similar processing times, the α-La yield was always significantly higher in the acidified samples of both species, in agreement with the results reported by other authors for HPP-treated bovine whey [34,35]. α-Lactalbumin yield for ovine and caprine samples was always higher than those reported for bovine concentrated native whey [35].

α-Lactalbumin purification degrees (Pur_α-La_) for caprine and ovine NWC_sup_ increased with the processing time, as has been described in the literature for HPP-processed cow whey [34,35] (Figure 6). At physiological pH, Pur_α-La_ in NWC_sup_ from HPP-treated samples was already higher than 80% for 2-min HPP treatments, and further significant improvements of this parameter could be obtained by HPP treatments for 4 min (goat) or 15 min (goat and sheep). On the contrary, Pur_α-La_ of acidified samples were always significantly lower than the corresponding samples treated at physiological pH, reaching values around 60% only after 15-min HPP treatment. These results are logical considering that β-Lg precipitation was significantly less affected by the HPP treatments in acidified samples, as previously stated, but again, they are clearly in contrast with what has been observed in bovine whey [34,35].

Our overall results for main process performance parameters with ovine and caprine NWCs confirmed that α-La is more tolerant to HPP treatments than β-Lg, which is in line with other studies carried out on bovine, ovine, and caprine milk and whey. This difference in baroresistance has been associated with structural differences between the molecules of whey proteins; α-La and β-Lg contain four and one intra-molecular disulfide bridges, respectively, and β-Lg has one free cysteine group [24,25,27,43,44]. It has been suggested that when high pressure is applied, β-Lg unfolds, exposing its free sulfhydryl group which is likely to interact with other molecules and form aggregates [27].

Regarding the effects of sample acidification, pH is a well-known factor affecting proteins’ structure. Bovine β-Lg has been reported to change its conformation when the pH is modified [45]. Loch et al. [46,47,48] studied the structure of ovine and caprine β-Lg, suggesting that they may suffer a similar structural change with acidic pH.

The combined action of HPP and acidification has been reported to increase β-Lg precipitation degree in bovine whey [34,35], probably because of β-Lg octamers’ formation [49] at pH 4.6; however, under our experimental conditions, we observed an opposite behavior for ovine and caprine β-Lg. After the HPP treatment, β-Lg was precipitated to a significantly less extent in acidified caprine and ovine NWCs than in those processed at physiological pH.

Some studies highlighted the strong homology between the ovine, caprine, and bovine WPs’ sequences [50]. Loch et al. [46,47,48] observed that WPs from different species exhibited different crystallization behaviors and interactions with an anionic exchange resin, suggesting that even small differences in the amino acid sequence could have a significant effect on the physicochemical properties of the protein, leading to a different response to factors influencing precipitation, such as acidification and HPP.

Under our chromatographic conditions we observed that ovine and caprine milk and whey samples contained one α-La variant, as frequently found in most of the breeds, and one or two β-Lg variants in goat and sheep samples, respectively, as other authors also reported [51,52,53,54] (Figure 7). The mean β-Lg/α-La ratio in control NWCs was in good agreement with values reported in the literature [54,55] (Appendix A).

Notwithstanding, we observed that both α-La and β-Lg variants in goat and sheep samples had shorter retention times than those of the corresponding bovine variants (Figure 7). This chromatographic behavior on a C18 stationary phase can ultimately be associated with a higher overall polarity of the goat and sheep WPs compared to the corresponding bovine molecules and is probably caused by small changes in the tertiary and quaternary structure of WPs in the three species. These differences could also explain the opposite effects of acidification and HPP on the β-Lg precipitation and α-La purification degrees observed in ovine and caprine samples compared to their bovine counterparts, as well as the different behaviors of caprine and ovine WPs as influenced by the HPP (Figure 4 and Figure 6) [35].

### 3.2. Hydrophobicity Index

The hydrophobicity index (H_I_) provides information about the quantity of hydrophobic groups on the surface of the protein. A decrease in H_I_ has been associated with the protein aggregation by hydrophobic bonds, while an increase may indicate the exposure of nonpolar amino acids on the molecule surface.

HPP treatments significantly decreased the overall hydrophobicity index (H_I_) in sheep and goat NWC_sup_ at physiological pH. This effect was already observed after 2-min HPP treatment, but longer processing times did not further reduced the H_I_ values (Figure 8). Acidification itself, without HPP treatment, also produced a significant reduction on the H_I_ of caprine and ovine samples.

However, H_I_ in the NWC_sup_ from acidified samples that were HPP-treated for 2 and 4 min were significantly higher in both species than the corresponding samples processed at physiological pH. In goat acidified samples, HPP for 2 and 4 min increased the initial H_I_ values, while longer processing times (15 min) decreased the H_I_ compared to the corresponding unprocessed (control) sample.

Nassar et al. [56] reported an increase in hydrophobicity in caprine micellar casein concentrate after HPP (400 MPa) at physiological pH. Similar results were also obtained by Altuner et al. [57] and Baier et al. [58], working with cow milk and whey proteins, respectively.

In a previous study, working under similar conditions with bovine NWC [35], we observed that H_I_ decreased more rapidly in NWC_sup_ in samples acidified before HPP treatment than in those processed at physiological pH.

In sheep and goat NWCs, acidification combined with HPP provoked almost opposite effects on H_I_ than those observed in bovine samples, which could be due to the above-mentioned structural differences of WPs in the three species, as well as to the higher residual concentration of β-Lg in acidified NWC_sup_.

### 3.3. Total Free Thiol Groups Index

The total free thiol groups index (TFT_I_) can indicate structural changes in the tertiary and quaternary protein conformation. A decrease in the TFT_I_ has been associated with the oxidation of the free sulfhydryl groups or the formation of new disulfide bonds, while an increase may suggest that the protein suffered a structural change that exposed the free sulfhydryl groups [59]. The TFT_I_ of untreated acidified samples was similar to the corresponding untreated samples at physiological pH of both sheep and goat species, indicating that acidification itself did not have a significant effect on this parameter (Figure 9).

On the contrary, HPP treatment significantly influenced TFT_I_ by decreasing its values in both sheep and goat samples. TFT_I_ decreased dramatically in sheep and goat samples that were HPP-treated at physiological pH, even for short processing times (Figure 9). However, when the NWC was acidified before HPP, the TFT_I_ decreased gradually, being comparable to the physiological one only after fifteen minutes of treatment.

Other authors have observed a similar effect of HPP on this parameter in bovine samples [60,61], while Romo et al. [35] have reported that sample acidification before HPP led to a higher reduction of TFTI in bovine NWC than the non-acidified variant. Since bovine β-Lg and caprine and ovine β-Lg had opposite responses to acidification before HPP, the decrease in TFT_I_ may be related to β-Lg denaturation, in agreement with Romo et al. [35].

As mentioned before for the Hi, this discrepancy observed between the effects of acidification and HPP in samples of different species could be related to structural variations in the WPs and help to explain the concomitant reduction of TFT_I_ and H_i_ with the β-Lg loss and α-La enrichment in the NWC_sup_.

## 4. Conclusions

As previously described in the case of bovine milk, whey, and NWCs, HPP has been demonstrated to be an effective technology to obtain α-La-enriched fractions from goat and sheep NWCs. The high tolerance of α-La to HPP was confirmed, while β-Lg was clearly affected by the HPP treatments.

Short HPP processing time (2 min) significantly decreased β-Lg concentration in the supernatant fraction after centrifugation (NWC_sup_), while concentration of α-La was unchanged. Longer HPP processing times (up to 15 min) progressively increased the α-La purification degree but also decreased the α-La yield.

The combined effects of sample acidification and the treatment were completely opposite to those reported in studies on bovine samples. Caprine and ovine NWCs treated at physiological pH provided better α-La yield, better α-La purification degree, and higher β-Lg precipitation degrees than the corresponding acidified samples, while the corresponding NWC_sup_ showed lower values for both H_I_ and TFT_I_, suggesting a higher extent of protein aggregation. Further studies would be desirable in order to better understand why small changes in the whey protein structure might endow this different behavior when HPP is applied.

## Figures and Tables

**Figure 1 foods-12-02688-f001:**
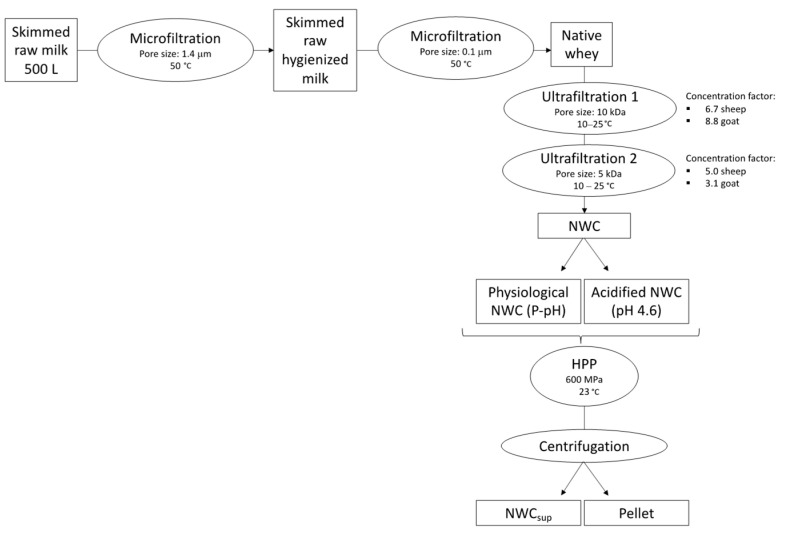
Workflow performed to obtain the α-La-enriched fraction from milk. After the membrane processes (micro and ultrafiltration), part of native whey concentrate (NWC) was HPP (high-pressure processing)-treated at physiological pH and another part at acidified pH. After centrifugation, two fractions, NWC supernatant (NWC_sup_) and pellet, were obtained.

**Figure 2 foods-12-02688-f002:**
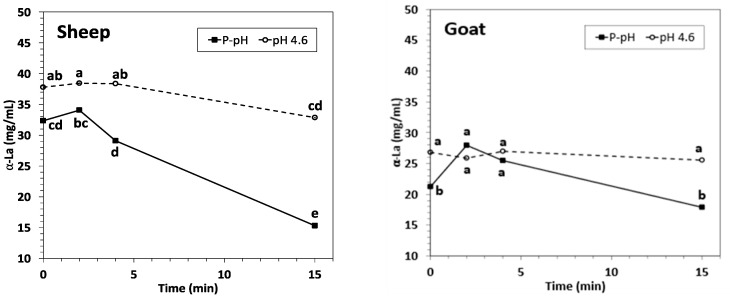
α-Lactalbumin concentration (mg/mL) in sheep (**left**) and goat (**right**) NWC_sup_ after HPP processing (600 MPa, 23 °C, 0–15 min). Different small letters in each graphic (sheep or goat) indicate significant differences between means (*p* < 0.05) according to Tukey’s test.

**Figure 3 foods-12-02688-f003:**
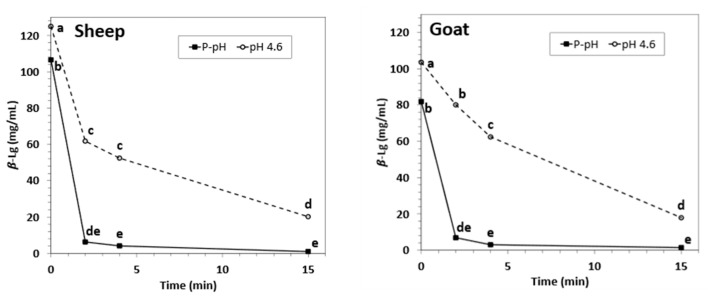
β-Lactoglobulin concentration (sum of A + B forms, mg/mL) in sheep (**left**) and goat (**right**) NWC_sup_ after HPP processing (600 MPa, 23 °C, 0–15 min). Different small letters in each graphic (sheep or goat) indicate significant differences between means (*p* < 0.05) according to Tukey’s test.

**Figure 4 foods-12-02688-f004:**
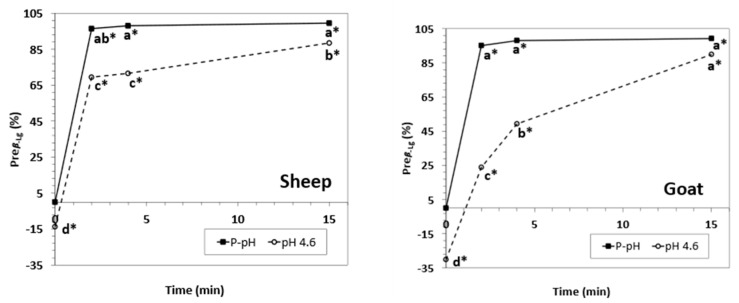
β-Lactoglobulin precipitation degree for sheep (**left**) and goat (**right**) NWC_sup_ after HPP processing (600 MPa, 23 °C, 0−15 min). Different small letters in each graphic (sheep or goat) indicate significant differences between means (*p* < 0.05) according to Tukey’s test. * Significant differences with untreated samples.

**Figure 5 foods-12-02688-f005:**
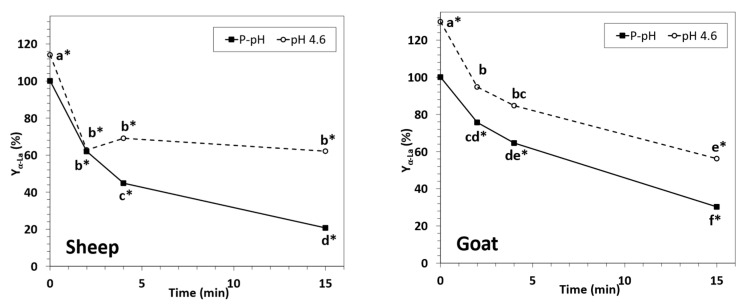
α-Lactalbumin yield for sheep (**left**) and goat (**right**) NWC_sup_ after HPP processing (600 MPa, 23 °C, 0–15 min). Different small letters in each graphic (sheep or goat) indicate significant differences between means (*p* < 0.05) according to Tukey’s test. * Significant differences with the respective untreated samples.

**Figure 6 foods-12-02688-f006:**
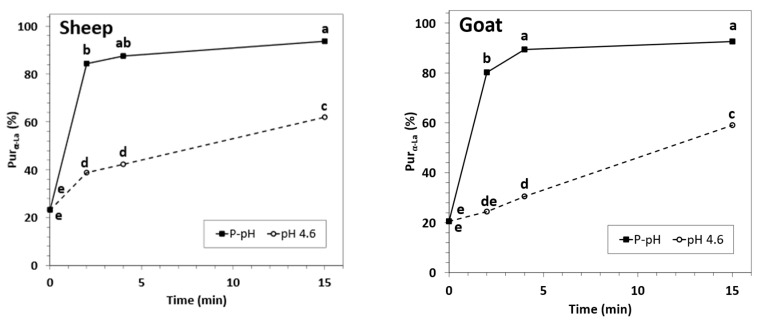
α-Lactalbumin purification degree for sheep (**left**) and goat (**right**) NWC_sup_ after HPP processing (600 MPa, 23 °C, 0–15 min). Different small letters in each graphic (sheep or goat) indicate significant differences between means (*p* < 0.05) according to Tukey’s test.

**Figure 7 foods-12-02688-f007:**
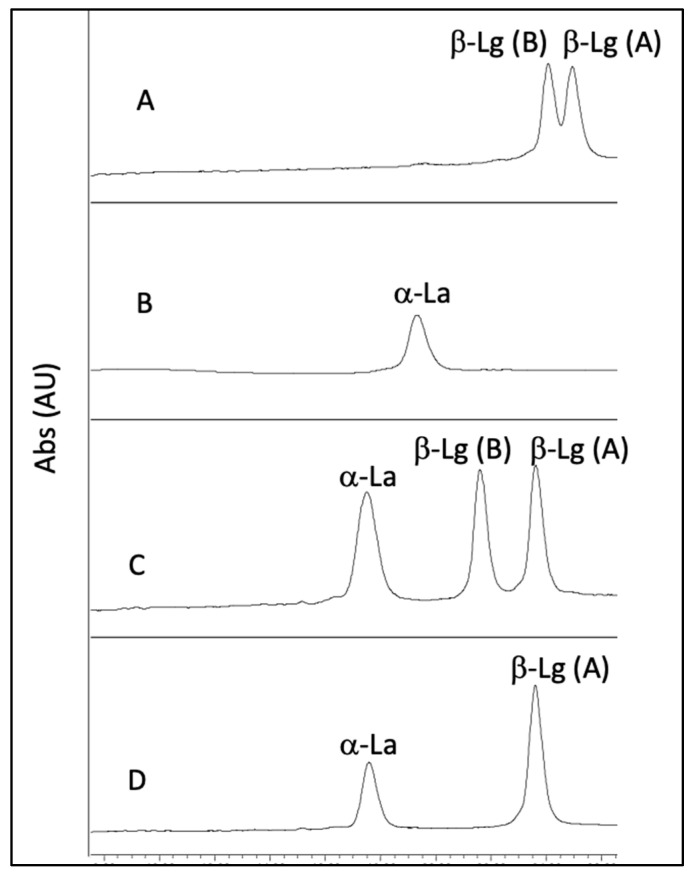
HPLC chromatographic separation of α-La and β-Lg variants in goat and sheep samples compared to a bovine WPs analyzed under the same conditions: **A** and **B** = standard solutions of bovine whey; **C** = sheep; **D** = goat.

**Figure 8 foods-12-02688-f008:**
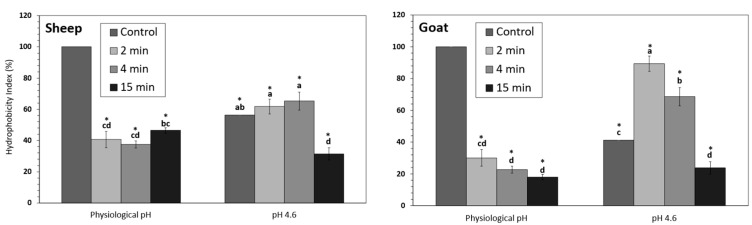
Hydrophobicity index (H_I_) in sheep (**left**) and goat (**right**) NWC_sup_ after HPP processing (600 MPa, 23 °C, 0–15 min). Different small letters in each graphic (sheep or goat) indicate significant differences between means (*p* < 0.05) according to Tukey’s test. * Significant differences with untreated samples.

**Figure 9 foods-12-02688-f009:**
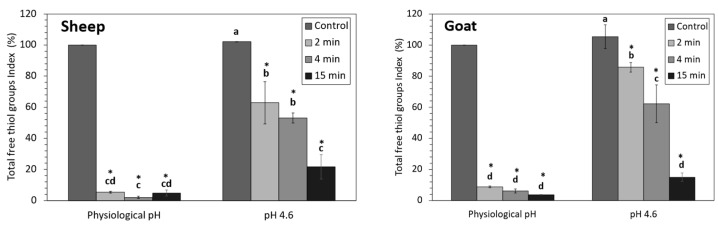
Total free thiol groups Index for sheep (**left**) and goat (**right**) NWC_sup_ after HPP processing (600 MPa, 23 °C, Control, 2, 5, and 15 min). Different small letters in each graphic (sheep or goat) indicate significant differences between means (*p* < 0.05) according to Tukey’s test. * Significant differences with untreated samples.

**Table 1 foods-12-02688-t001:** Composition of the native concentrate caprine and ovine whey before the treatment (% *w/w*).

Component	Goat Whey	Sheep Whey
Ash (% *w/w*)	0.60 ± 0.03	0.63 ± 0.04
Total dry matter (% *w/w*)	16.65 ± 2.26	20.60 ± 3.02
Lactose * (% *w/w*)	4.57 ± 0.33	4.00 ± 0.56
Fat (% *w/w*)	0.04 ± 0.02	0.01 ± 0.01
Protein (NT × 6.38) (%)	11.44 ± 1.79	15.96 ± 2.12
Physiological pH (P_pH_)	6.69 ± 0.05	6.74 ± 0.04
Concentration factor ^+^	27.28	33.5

* Calculated by subtracting from the total dry matter the total fat and protein and ash. ^+^ Calculated considering the protein concentrations before and after ultrafiltration.

## Data Availability

The data used to support the findings of this study can be made available by the corresponding author upon request.

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
