# Peer review of "Separation of α-Lactalbumin-Enriched Fractions from Caprine and Ovine Native Whey Concentrate by Combining Membrane and High-Pressure Processing"

_foods, 2023, doi:10.3390/foods12142688_

Round 1
Reviewer 1 Report
Specific comments:
Is there a difference between NWCs and NWCsup?
In the introduction please describe the differences between whey proteins of different species.
Figure 1: Give details/ parameters for MF and UF, like CF, temperature, membrane pore size.
Lines 242-246: ive te reason for differences between goat and sheep whey.
Lines 264-265: the concentration of beta-LG was reduced? In which fraction? supernatant?
Line 266: delete "and": performance parameters and are shown
Lines 298-301: What was the protein concentration in other studies? Please give more details on that, especially that you have pointed out that there is no studies on "concentrated native whey".
Lines 328-330: give possible reason for that.
The effect of pH only on whey protein is inadequately described. Why pH 4.6 was chosen?
Line 405: NWCs refer to supernatant?
Line 448: beta-LG
Lines 470-471: " sample acidification before HPP leaded to a 470 higher reduction of TFTI in bovine NWC" higher that what?
In general the quality of English language is fine. The sentences should not start with the abbreviations, i.e. beta-LG (i.e. line 290).
Author Response
Dear reviewer, thank you for your comments and suggestions.
Is there a difference between NWCs and NWCsup?
Yes, NCW stands for the whole native whey concentrate, before being separating into fractions (supernatant: NWCsup and pellet) by centrifugation after high pressure processing.
In the introduction please describe the differences between whey proteins of different species.
Short description has been added in line 50; more differences are deeper explained in the discussion (line 420-439).
Figure 1: Give details/ parameters for MF and UF, like CF, temperature, membrane pore size.
Modified. The global concentration factor added in Table 1 as well.
Lines 242-246: ive te reason for differences between goat and sheep whey.
Modified. A sentence was added in discussion (line 447) to suggest that the differences in protein profile/structure could explain the observed effect on the parameters considered in this study. On the other hand, deeper studies on the whey proteins’ differences between species was not the objective of the work.
Lines 264-265: the concentration of beta-LG was reduced? In which fraction? supernatant?
Yes, it was reduced in the supernatant. Sentence modified in the manuscript (line 288).
Line 266: delete "and": performance parameters and are shown
Modified.
Lines 298-301: What was the protein concentration in other studies? Please give more details on that, especially that you have pointed out that there is no studies on "concentrated native whey".
More details given. The studies cited in our paper were carried out on non-concentrated milk/whey (with the exception of our previous paper on bovine whey).
Lines 328-330: give possible reason for that.
See response to previous comment related to lines 242-246.
The effect of pH only on whey protein is inadequately described. Why pH 4.6 was chosen?
The effect of pH is described in Figures 2-6, 8 and 9 and discussed in lines 416-425. See also response to previous comment related to lines 242-246. pH 4.6 was chosen because it has been reported that promotes the formation of bovine beta-lactoglobulin, what could induce its precipitation when applying high pressure processing. Rewritten (line 422).
Line 405: NWCs refer to supernatant?
NCW stands for Native Whey Concentrate(s). So, in this case, it refers to the whole NWC for controls (untreated).
Line 448: beta-LG
Modified.
Lines 470-471: " sample acidification before HPP leaded to a 470 higher reduction of TFTI in bovine NWC" higher that what?
Rewritten in a clearer way.
In general the quality of English language is fine. The sentences should not start with the abbreviations, i.e. beta-LG (i.e. line 290).
Modified.
Reviewer 2 Report
The authors were able to explore the utilization of whey for 2 species and provide useful technical background information.
I would like the authors the clarify the labeling of the figures:
Figs 2,3,4,5,6,8,9: the letters showing significant differences were not able to clearly show the relationship between which 2 columns.
Author Response
Dear reviewer, thank you for your comments and suggestions.
Figs 2,3,4,5,6,8,9: the letters showing significant differences were not able to clearly show the relationship between which 2 columns.
Modified, sentence added to the legends in order to make it clearer.
Reviewer 3 Report
Manuscript ID: foods-2487039
In the manuscript submitted for comments, entitled “Separation of a-Lactalbumin Enriched Fractions from Caprine and Ovine Native Whey Concentrate by Combining Membrane and High-Pressure Processing”, the Authors studied the effect of high-pressure processing (HPP) on main proteins (a-lactalbumin and b-lactoglobulin) in ovine and caprine native whey concentrate (NWC) obtained by micro and ultrafiltration at pilot plant level without any heat treatment.
The paper, in line with the journal topic, is of interest to the scientific community, is well structured, scientifically supported, sound and logical, although I suggest some modifications to improve the impact of the research.
Below are my minor considerations line by line:
Lines 31: check key words: “goat, sheep, ewe”.
Line 52: check “[9].Unfortunately”.
Line 95-99 Fat content, protein content, total dry matter and ashes were determined according to Romo et al. [35] following the protocols ISO 1211/IDF1 [36], ISO 8968-3/IDF20-3 [37], ISO 2920:2004/IDF58:2004 [38] and BOE-A-1977-16116 [39], respectively. Lactose was estimated by subtracting the total fat, total protein, and ashes from the total dry matter, while pH was determined with a pH-meter (sensION+ PH3, HACH Co., Loveland, CO, USA).
I suggest the Authors insert the unit of measurement eg (% w/w) or (%).
Lines 102: table 1. I suggest the Authors to insert a short note about the lactose eg "Lactose was estimated by subtracting the total fat, total protein, and ashes from the total dry matter".
Line 227: “A T-test was performed”. Rewrite as “A t-test was performed ..”.
Figures 2 to 6, the letters indicating significance must be fully explained; furthermore, in some tables, the comparison of letters indicating significance is absent (e.g. in Figure 4, right, goat, control point 15 minutes).
Line 538: enter 'the date of access to the website'.
Minor editing of English language required.
Author Response
Dear reviewer, thank you for your comments and suggestions.
Lines 31: check key words: “goat, sheep, ewe”.
Checked. One word was eliminated (ewe).
Line 52: check “[9].Unfortunately”.
Modified.
Line 95-99 Fat content, protein content, total dry matter and ashes were determined according to Romo et al. [35] following the protocols ISO 1211/IDF1 [36], ISO 8968-3/IDF20-3 [37], ISO 2920:2004/IDF58:2004 [38] and BOE-A-1977-16116 [39], respectively. Lactose was estimated by subtracting the total fat, total protein, and ashes from the total dry matter, while pH was determined with a pH-meter (sensION+ PH3, HACH Co., Loveland, CO, USA).
I suggest the Authors insert the unit of measurement eg (% w/w) or (%).
Modified.
Lines 102: table 1. I suggest the Authors to insert a short note about the lactose eg "Lactose was estimated by subtracting the total fat, total protein, and ashes from the total dry matter".
Modified.
Line 227: “A T-test was performed”. Rewrite as “A t-test was performed ..”.
Modified.
Figures 2 to 6, the letters indicating significance must be fully explained; furthermore, in some tables, the comparison of letters indicating significance is absent (e.g. in Figure 4, rigth, goat, control point 15 minutes).
Figure legends have been modified and Figure 4 (goat, 15 min) improved.
The untreated controls for Figures 4, 5, 8 and 9 were considered as 100 % (Figures 5, 8 and 9) or 0 % (Figure 4) and they were used as the reference value for calculating the rest of the treatments. Therefore, these values (100% or 0%) were constant for untreated control samples, and they were not included in the global statistical analysis (ANOVA and Tukey test) to not underestimate the residual deviation. Therefore, the untreated controls for these Figures, do not have letters of significance .
However, we used the t-test with a 95% confidence interval for the mean value of each treatment to test if they significantly differed from the reference value (100% or 0%) of untreated samples. And those means that significantly differed from the reference values were marked with *
Line 538: enter 'the date of access to the website'.
Modified.